# Association between Sleep Quality and Physical Activity in Physical Education Students in Chile in the Pandemic Context: A Cross-Sectional Study

**DOI:** 10.3390/healthcare10101930

**Published:** 2022-10-01

**Authors:** Eugenio Merellano-Navarro, Natalia Bustamante-Ara, Javier Russell-Guzmán, Roberto Lagos-Hernández, Natasha Uribe, Andrés Godoy-Cumillaf

**Affiliations:** 1Department of Physical Activity Sciences, Faculty of Education Sciences, Universidad Católica del Maule, Talca 3530000, Chile; 2Grupo de Investigación en Educación Física, Salud y Calidad de Vida (EFISAL), Universidad Autónoma de Chile, Temuco 4780000, Chile

**Keywords:** physical activity, students, sleep quality, home confinement

## Abstract

**Background**: Due to the health restrictions put in place to face the pandemic, a decrease in the levels of physical activity and an alteration in the quality of sleep have been observed. One group susceptible to these changes is represented by students of physical education pedagogy, who, under normal conditions, present high levels of physical activity. However, the correlation between these variables has not been studied in the context of a pandemic lockdown. **Objective**: The aim of this study was to determine the prevalence of the quality of sleep and physical activity level as a function of gender and evaluate the association between quality of sleep and physical activity level in physical education pedagogy students during the COVID-19 pandemic considering sociodemographic and health characteristics. **Methods**: This was a cross-sectional study. An online questionnaire was applied to 280 Chilean university students of physical education pedagogy. The survey considered sociodemographic information, healthy habits, and self-reported health and included the International Physical Activity Questionnaire (IPAQ) for physical activity levels and the Pittsburgh Sleep Quality Index (PSQI) for sleep quality. **Results**: The prevalence of good sleep quality was 20.4%. Furthermore, 52.9% of students had a high level of physical activity. Regression analysis between PSQI global score and age-adjusted physical activity levels indicated that being male and presenting a high level of physical activity favored a better PSQI global score. **Conclusions**: The prevalence of good sleep quality was low in general and significantly lower in women. Being male and having a high level of physical activity during quarantine benefits good sleep quality, regardless of age.

## 1. Introduction

The coronavirus pandemic (COVID-19) has given rise to situations that are affecting the health of the population at all age levels. The rapid contagion caused a public health emergency of international interest declared at the end of January 2020 [1]. In South America, Chile was the fourth country—after Brazil, Ecuador, and Argentina—to disclose worrying contagion figures [2], which led to the enactment of a state of catastrophe [3] and a consequent reduction in social contact [4].

Confinement leads to reduced physical activity routines [5], fewer opportunities for sports activities [6], and increased anxiety and stress [5,7], and it may negatively affect sleep quality [7]. A recent study analyzed the increase in sleep problems during confinement [8] and suggested actions to prevent the associated problems [8,9].

Sleep helps restore vital functions for daily functioning, and sleep disturbance has an impact on general health [10]. This problem represents a high public health cost [11].

Poor sleep quality can be associated with mood disturbances and stress levels linked to pandemic living [12].

College students are a special group of people who are in a period of transition to adulthood, which can be a stressful time in a person’s life [13]. The impact of the pandemic has also affected higher education, transforming face-to-face teaching into a virtual modality, which has increased exposure to screens [14].

Pre-pandemic studies indicated a low percentage of good sleep quality in university students and that this condition is worse in women [15,16,17]. In this regard, logistic regression analysis has shown that sociodemographic variables, lifestyle, and medical problems are independent in explaining the differences between men and women [18]. Objective assessments of sleep by encephalographic activity indicate that women have an altered sleep architecture characterized by a shorter duration of stages N1 and N2 and a longer duration of stage N3 of the “non-rapid eye movement” (NREM) stage [19].

An Italian study analyzed changes in sleep habits in people in their early 40s, noting increased use of digital media close to bedtime and changes in sleep–wake rhythms (going to bed later and getting up later) [12]. Another Italian study conducted during the pandemic [20] analyzed changes in sleep, sense of time, and use of electronic devices in young adults, workers, and university students, showing that the latter presented more alterations in their bedtime, latency, and wake time, unlike young workers [20]. Other studies have reported associations between sleep quality and insufficient physical activity, both in young adults in Croatia and in university students in Saudi Arabia [7,21,22,23,24,25,26,27]. In the case of university students, this is characterized by high levels of sedentary behaviors and low levels of physical activity [28,29], which has increased due to the pandemic [30]. In this context, a study conducted during the first months of the pandemic showed high levels of poor sleep quality in the university population, and this variable was significantly higher in women [31,32].

A pre-pandemic study already pointed out that women are more likely to not get enough sleep compared to men due to factors such as screen time, the amount, level of intensity, and type of physical activity [33], and anxiety and stress that may have increased as a result of the sociodemographic conditions faced during the pandemic [34].

The practice of physical activity and adequate levels of physical fitness are essential to maintain good conditions of body functions during the pandemic [31]. This is relevant because some studies have indicated that there are preferences in the mode and type of physical activity performed according to gender, with women being the most affected since they prefer group activities, which were prohibited by health measures [35,36]. In the Chilean context, a study indicated that inactive university students with a more sedentary behavior during the pandemic reported lower well-being and worse mental health, as sedentary lifestyle is one of the variables that most affect mental health [37].

University students preparing to become physical education teachers are characterized by higher levels of physical fitness and physical activity, unlike their peers in other majors under normal conditions [38,39]. In a pandemic context, they could be a particularly susceptible group to a significant decrease in their physical activity levels. There are few studies that analyzed the prevalence and association between the quality of sleep and the level of physical activity in university students, and even fewer specific studies in physical education pedagogy students.

The study of this population allows us to identify some behaviors maintained during the pandemic in an active university population (physical education pedagogy students), in the framework the hypothesis that being physically active can be beneficial to improve sleep quality. The present study aimed (1) to determine the prevalence of the quality of sleep and physical activity level as a function of gender, and (2) to evaluate the association between sleep quality and physical activity level in physical education pedagogy students during the COVID-19 pandemic considering sociodemographic and health characteristics.

## 2. Methods

### 2.1. Setting and Design

The design of this study was cross-sectional. It was carried out virtually between July and December 2020 with students enrolled in the physical education pedagogy career at a university in Chile. This study adopted non-probabilistic convenience sampling for selecting participants. A total of 603 students aged 18 or older belonging to the three campuses where the career is taught (located in Santiago, Talca, and Temuco) were invited to participate. The inclusion criteria for the study were (a) being enrolled for the fall and/or spring 2020 semester in one of the three cities where the course is taught, and (b) having completed the academic year at the end of December 2020. Not completing the entire questionnaire online was considered an exclusion criterion (28 subjects did not answer all the questions in the questionnaires, preventing the analysis of the Pittsburg questionnaire and IPAQ). Of the 603 students invited to participate, 308 answered. The final sample consisted of 280 students (204 men and 76 women) who met the inclusion and exclusion criteria.

During that year, all universities in the country took on virtual teaching. The invitation to participate was made through internal social networks and institutional e-mails from which they could access the online survey (Google Forms), specifying the voluntary nature of their participation. The project had the approval of the Institutional Scientific Ethical Committee of the Universidad Autónoma de Chile (CEC-2320).

### 2.2. Instruments

Sociodemographic information and healthy habits: The survey included questions regarding their age, who they lived with, demographic zone, sports practice, type of physical activity, tobacco and alcohol consumption, presence of diseases, and family and personal history. This sociodemographic questionnaire was used in previous studies [40].

Health self-report: Questions were related to COVID-19 (physical symptoms within the last 14 days, tested positive for COVID-19, contact with an infected person, and use of health centers).

Self-reported physical activity: The International Physical Activity Questionnaire (IPAQ) was used. This instrument has an acceptable validity and reliability in its Spanish version [34,35]. The physical activity of the last 7 days was calculated by measuring the time spent in activity intensity by considering the estimated metabolic equivalent (MET) for that activity. Using IPAQ data processing and analysis guidelines, participants were classified into three categories: low, moderate, and high. For the high level, the criterion was to perform ≥ 3 days of vigorous activity and at least 1500 MET min-week, or to perform 7 days of a combination of walking, moderate-intensity, or vigorous-intensity activities achieving 3000 MET min-week. For the moderate level, it was to meet one of the following criteria: perform vigorous activity ≥ 3 days for ≥20 min, perform ≥ 5 days of moderate activity or walking for ≥ 30 min, or perform ≥5 days of any combination of activities with ≥600 MET min-week. For the low level, the classification criteria were those who did meet any of the criteria for either moderate or high levels of physical activity [41]. The IPAQ questionnaire has been validated in the adult population of different countries, including Chile. It presents acceptable validity (Spearman’s ρ = 0.30, 95% CI: 0.23–0.36) and reliability (Spearman’s ρ = 0.81, 95% CI: 0.79–0.82).

Sleep quality: The Pittsburgh Sleep Quality Index (PSQI) questionnaire was used [36]. This is a self-reported questionnaire that reports on sleep quality and signs of sleep disturbance during the 1 month period prior to completing the questionnaire. The Spanish version of the PSQI provides a reliable instrument with good validity [37,38,39]. This self-applicable questionnaire provides a global score of sleep quality through the evaluation of seven components: (1) subjective sleep quality, (2) sleep latency, (3) sleep duration, (4) habitual sleep efficiency, (5) sleep disturbances, (6) use of sleep medications, and (7) daytime dysfunction. The sum of the seven components creates a scale of 0–21 points (PSQI global score), with a higher score indicating worse sleep quality. As proposed by Buysse et al., good sleep quality was defined as a PSQI global score ≤ 5 and poor sleep quality was defined as a PSQI global score > 5 points. The PSQI has been widely applied in the general population, showing high sensitivity and specificity [42].

### 2.3. Statistical Analysis

Statistical analysis was performed using the IBM SPSS^®^ (version 24.0, IBM Corp., Armonk, NY, USA). The characteristics of the sample were obtained through a descriptive analysis using the mean ± standard deviation (SD) or percentage depending on the variable. The Mann–Whitney U test or Kruskal–Wallis Test was used if the variables were not normally distributed. To determine associations between categorical variables, the chi-square test was used. A binary logistic regression model was run to analyze the association between quality of sleep and physical activity level with sociodemographic and health variables (i.e., COVID-19 diagnosis, chronic illness, smoker, alcohol consumer, and mood); only significant variables were incorporated in the end model adjusted by sex. The significance level was set at *p* < 0.005.

## 3. Results

The characteristics of the participants are shown in Table 1. The sample consisted of 280 students with a response rate of 46%. The mean age was 21.4 ± 2.3 years, with a range between 18 and 29 years. A total of 27.1% of those evaluated were women. A total of 15.7% of the students resided in a rural area at the time of the survey. A total of 85% lived with both parents. In both indicators (i.e., residence in rural area and lived with both parents), there were no significant differences between men and women.

Regarding health status, only 3.2% of the sample had been diagnosed with COVID-19. Women reported having a disease three times more than men (*p* = 0.005) (i.e., diabetes, asthma, obesity, etc.). Women presented a 10.3% higher prevalence of tobacco consumption (*p* = 0.026) and a similar prevalence of alcohol consumption to men. More than 50% of the students reported a high level of physical activity. Specifically, men presented a 17% higher prevalence of high levels of physical activity than women; however, concerning the low level of physical activity, men and women presented similar prevalence. Physical activity levels and sex were significantly associated (*p* = 0.026). The global PSQI score was 8.4 ± 3.3 points, being higher in women (*p* = 0.002). The prevalence of good sleep quality was 20.4%, while poor sleep quality was 79.6%. A total of 23.5% of men had good sleep quality, which was 1.9 times higher than women (*p* = 0.031). Only 20.7% of participants slept more than 7 h per night as recommended; there was no difference between genders.

There was a positive association between the PSQI global score and the level of physical activity (*p* = 0.003) (Table 2). The students who had a high level of physical activity manifested better sleep quality. At the same time, it was determined that the prevalence of good sleep quality among students increased as the physical activity level improved, corresponding by 5% and 10% between the low to moderate level and the moderate to high level, respectively (*p* = 0.043). Figure 1 shows the PSQI score by physical activity level. A higher PSQI score denotes a worse indicator for each component. When investigating the relationship between each of the components of the PSQI global score and the level of physical activity, the association persisted for subjective quality, latency, duration, and sleep disturbance. Nevertheless, those who had good sleep quality did not report chronic illness and had higher prevalence of a good mood (*p* < 0.001); proportionally, there were fewer females (*p* = 0.018) than students with poor sleep quality (Figure 2).

Of the students with good sleep quality, 3.2% self-reported COVID-19 diagnosis and 9.5% were smokers, while two-thirds of the university students with poor sleep quality drank alcohol (*p* > 0.05).

Lastly, when performing a binary logistic regression analysis between sleep quality and physical activity level (adjusted for sex), a high level of physical activity 4.2-fold increased (OR = 4.2, 95% CI 1.1 to 15.7, *p* = 0.034) and good mood (no problems) 6.1-fold increased the probability of good sleep quality during the pandemic in physical education pedagogy students (OR = 6.1, 95% CI 3.2 to 11.6, *p* < 0.01).

## 4. Discussion

The main finding of the present study was that a high physical activity level was associated with a better quality of sleep in the physical education pedagogy students during the pandemic in 2020. A second finding was the low prevalence of good sleep quality in physical education pedagogy students (20.4%), being lower in women (11.8%), although, during the pandemic, 90% of this population followed the physical activity recommendations of the World Health Organization (WHO) for their age range.

Several studies have analyzed the prevalence and association between sleep quality and physical activity level in university students using the PSQI and IPAQ questionnaires, showing that physical activity is significantly associated with good sleep quality [23,24,32,39,40,41,42,43]. In contrast, specific studies in students of physical education pedagogy are scarcer. In this context, the work carried out in Romania in 2018 stands out, which demonstrated a correlation of average intensity between sleep quality and physical activity [24].

Our study reports that20.4% of university students of the physical education pedagogy career had good sleep quality, similar to the levels reported by Portilla-Maya et al. (21.2%) and Marelli et al. (26.2%). In general, the prevalence of university students with good sleep quality (PSQI global score ≤ 5) shows great variability in different countries, with values ranging from 21.2% to 74.1% [24,43,44,45,46]. Specifically, Badicu et al. observed a 74.1% prevalence of good sleep quality in physical education pedagogy students in a period before the COVID-19 pandemic, which contrasts radically with our results. Regarding the global PSQI score, in our study, it was 8.43 on average, the highest values close to the described range. Studies have reported values that vary from 5.9 to 9.2 points [42,44,45,46,47,48,49,50,51]. The variability in these results can be explained by the diversity of careers studied and their unique characteristics, as was the case with health careers, which have a higher prevalence of women and develop internships [32,48,49]. Therefore, it is suggested that the differences observed between our study and previous evidence may be due to multiple factors, such as the context of confinement, heterogeneity in physical activity levels, the virtual modality of teaching (classes and practices) in which the students were immersed, gender proportion in the sample, and the sociocultural differences between countries. Our results also show that 20.4% of the students were evaluated as having a good quality of sleep, with an overall PSQI score of 8.43. Longitudinal studies have determined the possible influence of the quarantine period associated with the COVID-19 pandemic on sleep quality in the adult population [50,51]. Alfonsi et al. reported higher scores on sleep latency, sleep efficiency, and medication use components during quarantine, with a delay in bedtime and wake-up times in Italian adults [52]. Regarding the effect of the COVID-19 pandemic on sleep quality in university students, the evidence is scarcer. A study on Italian students showed a significant increase in the global PSQI score, from a pre-quarantine value of 5.37 to a value of 6.97 during quarantine [18,53]. Likewise, a significant change was reported in the prevalence of good sleep quality, decreasing from 42% to 26.7% in the quarantine period [18]. On the other hand, the study by Marelli et al. reported that, during quarantine, there was a significant increase in sleep latency. In this study, sleep latency was the component that presented the highest average value in the students (1.5 ± 1.0) (data not shown).

The present study reported the existence of significant differences in favor of men in the quality of sleep in Chilean physical education students (1.9 times higher than that of women). These results are in accordance with what has been reported in the literature regarding university students of physical education pedagogy and other careers, where it has been observed that female students have lower sleep quality [24,43,45] and less sleep time [54]. Furthermore, these results are similar to the evidence involving different age groups [55,56,57,58]. On the other hand, evidence shows that women tend to have longer sleep latency and shorter sleep duration [59]. Although our results did not show significant differences by gender in the sleep latency and duration components, the PSQI global score showed a trend of higher average values for these scores in the female group. However, significant differences were detected in the components of subjective sleep quality, sleep disturbances, and daytime dysfunction at the gender level (data not shown). These differences observed in the quality of sleep according to gender can be caused by various factors, including hormonal changes inherent to the stages of the menstrual cycle [60] and the architecture of sleep of women [19], as well as factors related to the physical form of the person [60,61].

Regarding the analysis of the PSQI components, our results showed that the sleep latency component contributed the most to the global score obtained in our sample. This situation is consistent with that reported in a study carried out on university students, where it was shown that the sleep latency component is significantly higher in groups with poor sleep quality in relation to groups with good sleep quality [51]. In a study developed by Marelli et al., it was reported that, during quarantine, there was a significant increase in sleep latency. It was seen that poorer sleep quality has a negative impact on the learning, memory, and performance of students [62], which may affect their academic process. However, the evidence suggests that 1.5 latency in PSQI indicates a delay of between 16 and 30 min, which may not be that serious [63,64]. The distribution found in our study with respect to the Pittsburgh classification fits the evidence and reality of university students in other countries during periods of quarantine. However, these results should be interpreted carefully due to the multiple factors that could explain sleep quality in university students, such as academic demands, accessibility to virtual classes, and/or home comfort.

In relation to physical activity levels, our results indicated high levels of physical activity in the sample during the pandemic. These are similar results to those obtained by Zhang et al. [65], who found that the recommended doses of physical activity during this special period are higher than in previous studies to maintain physical and psychological health. In addition, our study found that sleep quality is associated with the high level of physical activity and good mood. A total of 2500 METs of weekly physical activity, equivalent to 80 min of moderate-intensity activity, appeared to minimize negative emotions during the COVID-19 pandemic. The types of physical activity performed were general and specific exercises of preference, such as high-intensity interval training or running, which facilitate metabolism, good sleep quality, and a calm mood.

Our sample shows an elevated prevalence of physical activity in the moderate and high levels (90.0%) and an association of physical activity levels as a function of sex (*p* = 0.026). This situation may be related to the nature of the career they are studying, as it involves a high practical component; therefore, students are probably more aware of the health benefits of regular physical activity [61,62,63,64,65,66,67,68,69].

The evaluation of physical activity is relevant in countries where restrictions on physical activity have been drastic, as is the case of Chile (prohibition of gyms, limited time slots for sports from 5 a.m. to 10 a.m. without considering the climatic diversity of the country, reduced capacity in sports practice facilities, and closure of parks) [70,71,72,73,74]. Some studies have shown that the restrictions implemented can influence habits that affect the short- and long-term health and wellbeing of students [75], such as variations in the schedules for the regular practice of physical activity and in the modality in which it is performed [22,76].

To date, few studies have analyzed the association between sleep quality and physical activity in the university population [37,38,39,40,41,42,77]. Our results indicate that, in university students, the quality of sleep improves as the level of physical activity increases, being more predominant in men. This gender gap can be explained by the sociocultural context and the opportunities for physical activity generated during the pandemic period. Men present a cultural affinity for outdoor activities (cycling, jogging, and trekking) and individual practice, as opposed to the group activities preferred by women, which have been limited during the confinement of the pandemic [78,79,80,81,82,83,84,85].

A study conducted on 440 university students in Saudi Arabia, of which 33% were from humanities majors, including the Faculty of Education, indicated that only 36.1% had a good sleep quality and 62.7% presented a prevalence of low levels of physical activity assessed through IPAQ, being more prevalent in women [37]. A study conducted prior to the pandemic by Štefan [40] on 2100 university students in Croatia indicated that those who did not comply with the physical activity recommendations had a 41% probability of having worse sleep quality considering sociodemographic, habit, and health adjustment variables (*p* < 0.01). Furthermore, a study conducted on 113 students of health sciences in India reported an association between sleep quality and physical activity, emphasizing the problems associated with the use of smartphones on physical activity and sleep quality of students [39].

However, evidence about sleep problems in the active population is scarce. Our results indicate that those students who presented a moderate level of physical activity had 15.1% of good sleep quality, increasing to 26.0% in those with a high level of physical activity. A study conducted in Norway with 98.8% of the active population, according to WHO recommendations, also showed an association between physical activity and sleep quality and stated that symptoms of anxiety and depression were associated with sleep problems [86]; the latter was also reported in a study conducted in Brazil [87]. A triad of physical activity, sleep, and mental health seems to emerge during the pandemic period, which is not always linked to a linear relationship; however, existing findings indicate that physical activity ameliorates symptoms of anxiety and depression in the population [23,88,89]. Our findings regarding the positive association between sleep quality and physical activity and the negative relationship with the PSQI global score agree with the results of a recent review conducted in 2021 by Memon et al. concerning university students [90].

Among the strengths of this study, we included a sample belonging to three important regions at the national level. Secondly, the cross-sectional design of our study allowed us to obtain data quickly without incurring major economic costs. There were also limitations of the study. Given the design of this study, the results cannot determine the cause–effect relationship. Due to the nature of the sample, the results cannot extrapolated to the entire population. Other variables that may influence sleep quality among university students such as stress, anxiety, depression, time in quarantine, or physiological variables (hormonal) were not considered. Regarding the results of the IPAQ and PSQI questionnaires, both address the perception of the participants in different extensions of time, which means that the results must be interpreted with caution. Lastly, our study presented an imbalance at the gender level; however, the sample studied is representative of the characteristics of the population studied. Future research should include an experimental design, based on exercise interventions, in order to understand the cause–effect relationship between physical activity and sleep quality in undergraduate physical education students in the pandemic context. In addition, we recommend including variables such as screen time, stress, and objective measurements of physical activity through accelerometry. This information could be useful for scientific communities interested in developing strategies that contribute to improving sleep quality in university students, mainly in women.

## 5. Conclusions

The present study analyzed sleep quality and physical activity in physical education pedagogy students during the COVID-19 pandemic period. Good sleep quality is associated with high levels of physical activity. The prevalence of good sleep quality was low and significantly lower in women. Being male and having a high level of physical activity during quarantine benefits good sleep quality, regardless of age. Maintaining healthy habits such as sleep and physical activity during the university period is relevant given the short- and long-term repercussions on health; moreover, there is a gender gap regarding both aspects, which must be considered in this population.

## Figures and Tables

**Figure 1 healthcare-10-01930-f001:**
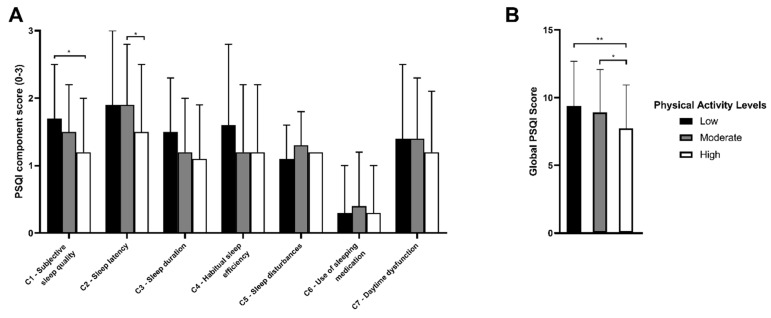
PSQI score according to physical activity level. (**A**) Mean PSQI component score according to IPAQ classification. (**B**) Mean global PSQI score according to IPAQ classification. Pittsburgh Sleep Quality Index; SE, standard error. Data are expressed as the mean with error bars. * *p* < 0.05; ** *p* < 0.01.

**Figure 2 healthcare-10-01930-f002:**
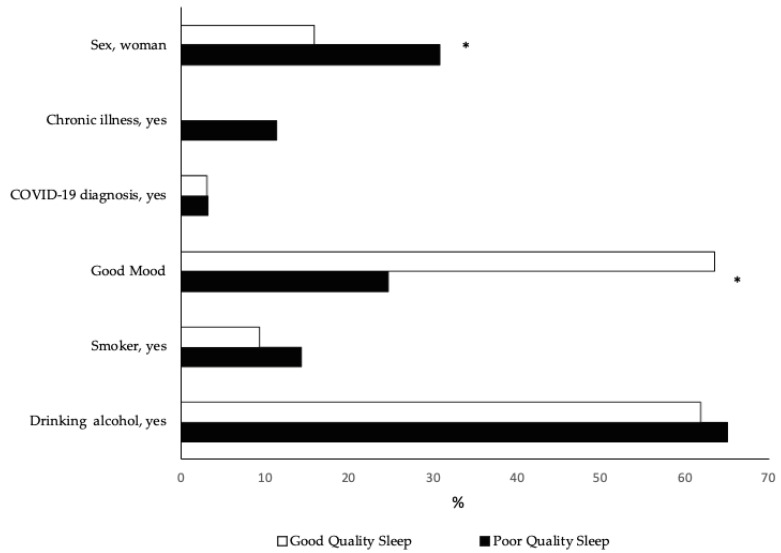
Sleep quality according to sociodemographic and health characteristics. * *p* < 0.05.

**Table 1 healthcare-10-01930-t001:** Characterization of the students.

	All Participants *n* = 280	Males *n* = 204	Females *n* = 76	*p*-Value
**Sex**		72.9	27.1%	
Age (years) mean ± SD	21.4 ± 2.3	21.4 ± 2.3	21.1 ± 2.2	0.155
Residence urban, n (%)	236 (84.3)	175 (85.8)	61 (80.3)	0.26
Academic year, n (%)				
≥4th year	67 (23.9)	48 (23.5)	19 (25.0)	0.24
3rd year	57 (20.4)	44 (21.6)	13 (17.1)	
2nd year	60 (21.4)	38 (18.6)	22 (28.9)	
1st year	96 (34.3)	74 (36.3)	22 (28.9)	
Living with, n (%)				0.079
Both parents	140 (50.0)	108 (52.9)	32 (42.1)	
Mother	98 (35.0)	71 (34.8)	27 (35.5)	
Friends or other	42 (15.0)	25 (12.3)	17 (22.4)	
Habits and Health, n (%)				
Chronic illness	23 (8.2)	11 (5.4)	12 (15.8)	**0.005**
COVID-19 diagnosis	9 (3.2)	7 (3.4)	2 (2.6)	0.736
Smoker	38 (13.6)	22 (10.8)	16 (21.1)	**0.026**
Drinking alcohol	179 (63.9)	130 (63.7)	49 (64.5)	0.908
Mood				
Good (without problems)	100 (32.7)	79 (35.7)	21 (24.7)	0.065
Bad (with mild or severe problems)	206 (67.3)	142 (64.3)	64 (75.3)	
Physical Activity Level, IPAQ, n (%)				
High	146 (52.1)	116 (56.9)	30 (39.5)	**0.026**
Moderate	106 (37.9)	68 (33.3)	38 (50.0)	
Low	28 (10.0)	20 (9.8)	8 (10.5)	
Met guidelines WHO *, n (%)				
Yes	252 (90.0)	184 (90.2)	68 (89.5)	0.858
Type of physical activity on pandemic, n (%)				
General	138 (49.3)	102 (50.0)	36 (47.4)	0.465
Specific	124 (44.3)	87 (42.6)	37 (48.7)	
None	18 (6.4)	15 (7.4)	3 (3.9)	
PSQI global score, mean ± DS	8.4 ± 3.3	8.0 ± 3.1	9.3 ± 3.4	**0.039 ***
Good Sleep Quality, n (%)	57 (20.4)	48 (23.5)	9 (11.8)	**0.031**
Poor Sleep Quality, n (%)	223 (79.6)	156 (76.5)	67 (88.2)	
Sleep duration, n (%)				
<7 h	222 (79.3)	158 (77.5)	64 (84.2)	0.215
>7 h	58 (20.7)	46 (22.5)	12 (15.8)	

IPAQ, International Physical Activity Questionnaire; IFIS, International Fitness Scale; PSQI, Pittsburgh Sleep Quality Index; SD, standard deviation. * Met guidelines WHO: Participants perform at least 150 min/week of MPA, 75 min/week of VPA, or an equivalent combination greater than 600 METS/min/week of MVPA. * Analysis carried out using Mann–Whitney U test. The values in bold indicate a statistical significance for *p* < 0.05.

**Table 2 healthcare-10-01930-t002:** Physical activity level and quality of sleep.

	High *n* (%)	Moderate *n* (%)	Low *n* (%)	*p*-Value
PSQI global score, mean ±SD	7.7 ± 3.2	8.9 ± 3.2	9.4 ± 3.3	0.003 *
Good sleep quality	38 (26.0)	16 (15.1)	3 (10.7)	0.043
Poor sleep quality	108 (74.0)	90 (84.9)	25 (89.3)	
Sleep duration				
<7 h	109 (74.7)	87 (82.1)	26 (92.9)	0.062
>7 h	37 (25.3)	19 (17.9)	2 (7.1)	

PSQI, Pittsburgh Sleep Quality Index. * Analysis carried out using Kruskal–Wallis Test.

## Data Availability

The data presented in this study are available on request from the corresponding author.

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
