# Peer review of "Association between Sleep Quality and Physical Activity in Physical Education Students in Chile in the Pandemic Context: A Cross-Sectional Study"

_healthcare, 2022, doi:10.3390/healthcare10101930_

Round 1
Reviewer 1 Report
The quality of the article is good from a scientific point of view, but there is a basic question about this study that the authors should answer:
This result can be achieved without conducting a study, so what was the need to conduct this study?
The article needs to be edited for better presentation.
Regards

Author Response
Reviewer 1
The quality of the article is good from a scientific point of view, but there are ambiguities
that should be resolved in the article.
- General purpose in the final part of the introduction: The general result of this study is clear, but the relationships considered with regard to demographic characteristics are important. It is necessary to address this issue in detail. Therefore, it is necessary for the authors to explain more about the innovation of this work?
R: Thank you very much for the observation. This will undoubtedly contribute to enhance the article. We have added two paragraphs in the introduction with the requested comment.
Methods:
- 2.1. Setting and Design: Please explain what was the reason for choosing 280 students
from 308?
R: The exclusion criterion (page 3, line 113) was not completing the entire questionnaire. Therefore, we reduced from 308 subjects to 280.
- 2.2. Instruments: Mention the validity and reliability of the questionnaires used?
R: This information is incorporated in the instruments made (page XX).
- Explain the instructions of the IPAQ questionnaire regarding the division into three categories of low, medium and high in this part as well?
R: This information is included in the methodology (page XXX).
- The analysis of the results is well done.
R: Thanks
- Use the references of 2022 in the discussion of the article.
R: 2021 and 2022 quotations were incorporated in the introduction and discussion.
- The whole article should be edited in terms of native English. Minor writing errors can be seen in the article, such as:
R: All English corrections were made. In addition, the entire document was reviewed to verify the quality of the English.

Reviewer 2 Report
This manuscript examined the sleep quality and physical activity and their relationship in Physical Education students. The study is of interest to readers.
The Abstract presented was clear except for the Conclusion statement, which did not echo the key objectives of the study but reported findings of minor objectives of the study. The Introduction read well up until Line 52, after which re-organisation of the content is needed (please see specific comments in the manuscript). The Methods section also requires some re-arrangement so that similar contents are grouped accordingly. The Result section requires much revision. Some issues include:
- Table 3 is missing
- Figure 2 can be replaced with description of the data
- It would be of value to add a Table that displays the findings of the regression analysis
- Clarify what the sleep components mean for Figure 1A
- Need depict between group differences for statistically significant data for Table 2, e.g., whether significance was between High and Low group or between Moderate and Low group.
- Re-write sentences for clarity and readability.
The authors need to structure the Result sections clearly, for example, reporting first on socio-demographics (everything related to this topic need to come under this section), followed by reporting on sleep quality (and any differences for males and females), then on physical activity levels (and for any differences between males and females), then finally on the correlations/relationship between sleep quality and physical activity levels.
The Discussion also requires revisions, and the order of discussions could follow that presented in the Result section to present a logical flow of contents. The authors are encouraged to reiterate their findings first up before discussing their findings with supporting literature (that agree/disagree with their findings). This manner of presentation allows emphasis of their findings and allow the readers to see the importance of their findings.
The authors are requested to describe findings of the literature specifically, for an example, lines 211-213: “Several studies have analyzed the prevalence and association between sleep quality and physical activity level in university students using the PSQI and IPAQ questionnaires [23,24,39–42].”– This sentence in its present form is not informative. Therefore, need add specific information such as “positive or negative associations” between PSQI and IPAQ scores and therefore value-added.
English language and expressions: The Abstract and part of the Introduction were very well written, but this cannot be said for the Results and Discussion sections. Sentences tend to be too long, and some require clarifications as the messages were ambiguous. It is suggested that the authors pay attention to these details.
Please see specific comments in the manuscript and kindly address them.

Author Response
Reviewer 2
- Line 224-25: For clarity and readability, please re-write this sentence
R: Done
- Line 28-29: The Conclusion appears incongruent with the key objectives of the study. Please re-write.
R: Done
- Line 56-57: "phases " is an incorrect term applied to polysomnography. Please use N1 and N2 for NREM sleep stages 1 and 2. In the modified nomenclature for sleep, Stage 4 is no longer reported. In sleep scoring Stage 3&4 have been combined to the new terminology of N3 (slow wave sleep).
R: Thanks for the info. Modification made.
- Line 59-64: these studies were about sleep quality reported during the pandemic - please say so.
R: Done.
- Since the focus of this study is on the relationship between physical activity and sleep quality, it is pertinent for the authors to provide an overview description of the published work (to get a feel for the literature as you have indicated in the Discussion with the citations [23,24,39–42].
R: Done.
- Line 67: please clarify in what context is "All of the above is relevant"?
R: Done.
- The section on the relationship between sleep quality and physical activity (Line 66-74) should be shifted down to after the description of 'physical activity levels' in Uni students, after around Line 81. There is insufficient description about the association between sleep quality and physical activity. Please add more details.
R: Done.
- This section needs re-organisation, for example, the statements on participant chracteristics and those on inclusion/exclusion criteria need to be grouped together, then followed by the paragraph on Ethics approval.
R: Done.
- For the IPAQ and PSQI, please indicate the reference period (time frame) for the questionnaires.
R: All questionnaires were administered during the months of July to December 2020 through an online questionnaire.
- In the Discussion section, please comment on any impact on the interpretation of the questionnaire in relation to the reference period; the IPAQ asked about PA during the past week, whereas the PSQI asked about SQ during the past month.
R: Thanks for the comment. Information added in discussion.
- Lines 143-144: please change "was composed of" to "consisted of" and change "students, obtaining a response rate" to "students with a response rate"
R: Done.
- Line 146-148: please re-write this sentence or split the sentence into two since the last sentence ("and there were no differences between men and women in these indicators") causes confusion in its meaning.
R: Done.
- Line 147: Please rephrase "your mother". What about with father?
R: Modified.
- Line 148-152: Whilst the content of this sentence can be understood, it's quite messy. Please break it up into two sentences for ease of reading and clarity.
R: Done.
- Line 150: change 'prevalence' to 'prevalent
R: Done.
- Line 152: “followed of”?? Please clarify.
R: Done.
- Line 156: Please clarify how (in relation to analysis) and where "strength" data were derived from.
R: Done.
- Line 170: Please clarify what “it refers to”.
R: Done
- Line 170-173: kindly re-write sentence to 173 improve clarity.
R: Done
- Line 169 and Line 173: Can authors please report the results for Table 2 first before that for Table 3. Alternatively, please re-arrange the Tables.
It is more logical to present table data chronologically. Table 3 is missing! Suggest presenting a figure for the correlational data instead of Table 3.
R: The results were reorganized.
- For Table 2, please indicate which group differences were depicted by the significant p-level and whether at p<0.05 or at other p values.
R: Table 1 shows how physical activity levels are associated with gender (p=0.026). The p-value is not exclusive to the moderate category. This result was included in the discussion.
- Table 1: The 'Moderate' category for IPAQ was statistically significant - was this finding mentioned in the Results and discussed?
R: This information was added to the results.
- Figure 1A, For clarity, for the various components, please explain what the PSQI component score means, for example, does a lower score always represent "no sleeping difficulties" whether it's about sleep onset latency or sleep duration?
R: This information is added to the results:
Figure 1 shows PSQI score by physical activity level. The higher the PSQI component score PSQI, the worse the indicator for each component. When investigating the relationship between each of the components of the PSQI global score and the level of physical activity, the association persisted for subjective quality, latency, duration, and sleep disturbance.
- Figure 1B: please add "score" to IPAQ - it's about the scores, not the questionnaire.
R: The data presented is not the IPAQ score. it is the level of physical activity categorized by IPAQ. This categorization is based on the number of METS per week. Included in Figure B is the description of physical activity levels.
- Line 184: please mention that the error bars were standard error. SE was not presented anywhere in the figure.
R: Done. Included in the table description: Data expressed as mean and error bars. * p < 0.05; ** p < 0.01.
- Line 186: please use lower case for Good Quality Sleep. Lines 186-190 were a repetition of Lines 171-173. Information in Lines 189-190 should be incorporated into the earlier section (lines 171-173.
R: Done.
- Figure 2 about sociodemographics and health data should be presented first, ie., as Figure 1 before presenting the data about PSQI and IPAQ scores. In addition, in this figure, need be systematic in your presentation of the data, ie, demongraphic factors first, followed by health data (e.g., move "sex" up to the top of the figure)
R: The error has been corrected. The results section presents the information in order.
- instead of "good quality sleep" why not "good sleepers" and "poor sleepers"? Please indicate p-value significant levels.
R: The p-value is expressed with *. Only Sex/female and good mood show statistically significant differences.
- Lines 195-200: please re-write this sentence for clarity!
R: The paragraph was rewritten.
- Lines 195-200: Given the importance of this data 200 (presented as a key finding in the Abstract, a table of your findings should be presented.
R: We appreciate your comment, however, we consider that it is not necessary to present these results in a new table.
- Line 202: Please clarify where the correlational data for good mood and sleep quality were presented
R: The phrase good mood was removed.
- Lines 209-210: please cite a reference
R: Done.
- Line 210-211: Kindly clarify the message for this sentence.
R: Done.
- Line 211-213: Please indicate what type of prevalence/ associations were observed - authors have been very ambicuous with data reporting.
R: Done.
- Line 215: What were the findings of the Romania study?
R: Done.
- Lines 216-232: please discuss your own findings first up and then refer to other published work to make your work relevant in this Discussion. I suggest re-writing this whole section.
R: Thank you for your comment. This section has been rewritten.
- Lines 233-237: Need talk about your own findings on the sleep components and then discuss these findings with reference to published studies
R: Done.
- Line 238-239: Need reiterate the finding of gender difference found in this study to give contest before discussing the details. "Why "for the
first time, if your data were in accordance with other studies (Lines 239-240)
R: Thank you for your comment. The request was made and the concept was eliminated the “first time”.
- Line 243: inappropriate use of the term "even consistent"
R: Was modified.
- Lines 250-521: there may be other factors to explain why
female student slept worse, e.g. screen time, social activities etc Please re-evaluate and discuss sex difference in your study.
R: Thanks for the comment. They added factors that could influence the differences.
- Lines 252-265 - the discussion of this section is good. But would suggest the authors start the discussion by first reporting own data (ie lines 261-2). Then jump into discussion of other literature to give context.
R: Suggested was done.
- Line 266: Sleep onset latency of 1.5 in the PSQI indicates a delay of between 16-30 minutes. It's not that severe. Please clarify the message intended for the reader.
R: Thanks for the information. Clarified message.
- Line 271, please provide examples of factors that interfere with sleep in this group of students.
R: Examples are included. The paragraph is reworded as follows: . However, these results should be interpreted with caution because of the multiple factors that could explain sleep quality in college students, such as academic demands, accessibility to virtual classes, and/or home comforts.
- Lines 273-277: again, please report own data first, ie "your results agree with the published literature" rather than "published data agree with your data" so to give emphasis to your findings
R: The entire paragraph was rewritten.
- This section (lines 294-301) contains discussion on associations between SQ and PA and gender gap on PA. But the authors had started a discussion on the association back in Line 209. Need re-arrange these discussions appropriately.
R: Thank you for your comments. The gender gaps are mentioned in the first paragraph of the discussion in order to introduce the main findings, which are developed later in the discussion.
- Conclusion section:
The study title addresses the association between SQ and PA. The author's conclusion did not mention the outcome of this relationship.
R: Thank you for your comments. We add in the conclusion that quality sleep is associated with high levels of physical activity.

Round 2
Reviewer 2 Report
In its present form (without English editing), the manuscript is not publishable. We recommend that the authors seek help with English editing, as clarification is required with many of the sentences in the manuscript (only a few have been listed in the specific comments below).
Line 64: please re-write this sentence for clarity.
Line 65: please spell out what the "distractors" are. It is not clear what is meant by the distractors.
Line 65: Incorrect use of the term "motivated" - please change it to an appropriate term.
Line 80: please change the phrase “to determine the prevalence of sleep quality” to “to determine the prevalence of the quality of sleep”
Line 91: It is not clear why “Completing the entire questionnaire online was considered an exclusion criterion”. Please explain.
Line 100: past tense “lived”
Line 104: For the IPAQ and PSQI, please indicate the reference period (time frame) for the questionnaires, for example, “the IPAQ asked about the time spent being physically active in the last 7 days”. “The PSQI asked about usual sleep habits during the past month only”.
In the Discussion section, please comment on any impact on the interpretation of the
questionnaire in relation to the reference period; the IPAQ asked about physical activity during the past week, whereas the PSQI asked about sleep quality during the past month.
Line 135: please use past tense for “live”
Line 136: “In both indicators there were no significant differences between men and women” – Please clarify what does “indicators” mean? What does it refer to?
Line 137: “Women reported having some disease 3 times more than men” – Please clarify what “some diseases” were? Very unclear.
Line 139-142: require English editing.
Line 183-185: “In general, the relationship between physical activity and sleep in adults and older adults is well established [22,23]. However, due to the diversity of instruments that have been used, these estimates do not allow generating a consensus in young adults [43].” These two sentences are difficult to follow: 1) it was not clear what the relationship was between physical activity and sleep. Was it a positive or negative one. The authors need to state this. 2) The second sentence: what were “estimates” refer to?? Consensus in what??
I suggest deleting these two sentences (line 183-185), since the sentence that follows (line 185-187) spells out the relationship anyway.
Line 188-189: Please re-write this sentence “In this context, the work carried out in Romania in 2018 stands out, where a 188 correlation of average intensity between sleep quality and physical activity is demonstrated.” Two issues: the term “stands out” and “average intensity”
There are two sections (line 202-205 AND line 228-236) that addressed “sleep latency”. Please integrate these two paragraphs.
Suggest integrate the paragraph (line 218-227) into the paragraph (line 190-201) because both paragraphs address “sleep quality”.
Suggest move down the paragraph (line 183-189) and integrate into the paragraph (line 255-268) to improve flow of contents.
Line 295: “Sleep quality is associated with high levels of physical activity.” Please add “good” sleep quality is associated with…
Author Response
Reviewer 2
Comments and Suggestions for Authors
- In its present form (without English editing), the manuscript is not publishable. We recommend that the authors seek help with English editing, as clarification is required with many of the sentences in the manuscript (only a few have been listed in the specific comments below).
R: The manuscript was edited by an English specialist
- Line 64: please re-write this sentence for clarity.
R: Done
- Line 65: please spell out what the "distractors" are. It is not clear what is meant by the distractors.
R: Done
- Line 65: Incorrect use of the term "motivated" - please change it to an appropriate term.
R: Done
- Line 80: please change the phrase “to determine the prevalence of sleep quality” to “to determine the prevalence of the quality of sleep”
R: Done
- Line 91: It is not clear why “Completing the entire questionnaire online was considered an exclusion criterion”. Please explain.
R: Done
- Line 100: past tense “lived”
R: Done
- Line 104: For the IPAQ and PSQI, please indicate the reference period (time frame) for the questionnaires, for example, “the IPAQ asked about the time spent being physically active in the last 7 days”. “The PSQI asked about usual sleep habits during the past month only”.
R: Done
- In the Discussion section, please comment on any impact on the interpretation of the
questionnaire in relation to the reference period; the IPAQ asked about physical activity during the past week, whereas the PSQI asked about sleep quality during the past month.
R: Done
- Line 135: please use past tense for “live”
R: Done
- Line 136: “In both indicators there were no significant differences between men and women” – Please clarify what does “indicators” mean? What does it refer to?
R: Done
- Line 137: “Women reported having some disease 3 times more than men” – Please clarify what “some diseases” were? Very unclear.
R: Done
- Line 139-142: require English editing.
R: Done
- Line 183-185: “In general, the relationship between physical activity and sleep in adults and older adults is well established [22,23]. However, due to the diversity of instruments that have been used, these estimates do not allow generating a consensus in young adults [43].” These two sentences are difficult to follow: 1) it was not clear what the relationship was between physical activity and sleep. Was it a positive or negative one. The authors need to state this. 2) The second sentence: what were “estimates” refer to?? Consensus in what??
R: Done.
- I suggest deleting these two sentences (line 183-185), since the sentence that follows (line 185-187) spells out the relationship anyway.
R: Done
- Line 188-189: Please re-write this sentence “In this context, the work carried out in Romania in 2018 stands out, where a 188 correlation of average intensity between sleep quality and physical activity is demonstrated.” Two issues: the term “stands out” and “average intensity”
R: Done
- There are two sections (line 202-205 AND line 228-236) that addressed “sleep latency”. Please integrate these two paragraphs.
R: Done.
- Suggest integrate the paragraph (line 218-227) into the paragraph (line 190-201) because both paragraphs address “sleep quality”.
R: Done
- Suggest move down the paragraph (line 183-189) and integrate into the paragraph (line 255-268) to improve flow of contents.
R: Done
- Line 295: “Sleep quality is associated with high levels of physical activity.” Please add “good” sleep quality is associated with…
R: Done
